# Modeling the dynamic behavior of a droplet evaporation device for the delivery of isotopically calibrated low-humidity water vapor

Erik Kerstel[1]

[1]Univ. Grenoble Alpes, CNRS, LIPhy, 38000 Grenoble, France

**Correspondence:** Erik Kerstel (erik.kerstel@univ-grenoble-alpes.fr)

**Abstract.** A model is presented that gives a quantitative description of the dynamic behavior in terms of water vapor concentration (humidity) and isotope ratios of a low-humidity water vapor generator. The generator is based on the evaporation of a nanoliter-droplet produced at the end of a syringe needle by balancing the inlet water flow and the evaporation of water from the droplet surface into a dry air stream. The humidity level is adjusted by changing the speed of the high-precision syringe pump and, if needed, the dry air flow. The generator was developed specifically for use with laser-based water isotope analyzers in Antarctica, and recently described in Leroy-Dos Santos et al. (2020). Apart from operating parameters such as temperature, pressure, water and dry air flows, the model has as "free" input parameters the water isotope fractionation factors and the evaporation rate. We show that the experimental data constrain these parameters to physically realistic values that are in reasonable to good agreement with literature values where available. With the advent of new ultra-precise isotope ratio spectrometers the approach used here may permit measuring not only the evaporation rate, but also the effective fractionation factors and isotopologue dependent diffusivity ratios, in the evaporation of small droplets.

## 1 Introduction

Water is arguably the most important molecule in Earth's atmosphere. The large enthalpy change associated with the evaporation and condensation of water causes it to dominate the global redistribution of energy by tropospheric transport of latent heat. Water vapor is also the most important greenhouse gas. The natural atmospheric greenhouse effect warms Earth's surface by 33 K to hospitable temperatures of on average 15 °C. About 75% of this temperature increase is generated by water vapor and clouds, as a feedback effect driven by the non-condensable greenhouse agents, and foremost carbon dioxide (Lacis et al., 2010). This feedback effect, in turn, is a superposition of a multitude of large and (especially as clouds are involved) complex individual processes that partially cancel each other. Due to this complexity, water, in the form of water vapor, as well as liquid and crystal phase water inside clouds, is by far the largest unknown in current climate models (IPCC, 2013). Atmospheric data of relevant tracers, which may help to disentangle and quantify the many relevant processes, are desperately needed. Of these, the isotopic composition of water (in particular the ratios $^2H/^1H$ and $^{18}O/^{16}O$, but also $^{17}O/^{16}O$ and the derived quantities of deuterium- and $^{17}O$-excess) is arguably the best candidate, as all processes in which water is involved are isotope-dependent. Therefore, water isotope ratios enable identification of different moist air masses and following of their mixing; they also re-

flect the evaporation and condensation history of the moist air in question. In the journal Nature, the climate researcher Gavin Schmidt actually called the water isotopes "the most super-duper fantastic thing ever" (Tollefson, 2008).

It is also no overstatement to say that laser-based isotope analyzers have revolutionized the field of water isotope ratio instrumentation, until not so long ago dominated by Isotope Ratio Mass Spectrometers (e.g., Kerstel, 2004; Kerstel and Gianfrani, 2008). In particular, laser instruments have enabled continuous measurements of low-humidity atmospheric air in airborne and Antarctic field settings (see, among others, Iannone et al., 2009b, 2010; Moyer et al., 2013; Steen-Larsen et al., 2013; Casado et al., 2016; Ritter et al., 2016; Bréant et al., 2019). In order to calibrate such instruments against international standard and reference materials that are all in liquid form, it is necessary to bring these into the vapor phase without causing fractionation, or alternatively with well-controlled, quantitative fractionation, while also controlling the level of humidity (the volume mixing ratio). Several solutions have been proposed and developed into prototypes and commercial instruments, but few are capable of delivering a stable supply at low humidity levels (Iannone et al., 2009a; Sturm and Knohl, 2010; Gkinis et al., 2010; Tremoy et al., 2011). One approach is that of the instrument developed in our laboratory, based on nanoliter- (nL-) sized droplet evaporation, with the specific aim of calibrating laser-based isotope ratio ($^2$H/$^1$H, $^{17}$O/$^{16}$O, and $^{18}$O/$^{16}$O in water) analyzers deployed in Antarctica and first reported in Landsberg et al. (2014). This prototype instrument has undergone significant engineering developments in order to improve its performance and robustness, as reported in Leroy-Dos Santos et al. (2020).

The current paper describes a theoretical model of the droplet evaporation that was developed to quantitatively describe the operation of the device. The model is presented here in detail and subsequently applied to data collected with the original prototype, as this device allowed us to easily modify some crucial parameters (such as the velocity of the air in the evaporation chamber) and, being equipped with two instead of one syringe pump, enabled a rapid switching between two independently prepared humid air flows. It also showed non-ideal behavior that was eliminated in the final version, but that enabled a more extensive test of the model. Finally, whereas it was deemed sufficient for the new instrument to be passively temperature stabilized to 20 ±1 °C, the prototype instrument had its evaporation chamber actively stabilized at 35.0 ±0.1 °C.

The theoretical understanding of the dynamic behavior has enabled the identification of the droplet evaporation device as an independent tool to investigate isotope fractionation factors involved in liquid-vapor transitions, as well as isotope fractionation occurring during the process of evaporation of cloud water droplets. The same is true for the determination of the evaporation rate of nano- and micro-liter-sized droplets, which has been the subject of a large body of research, starting with the fundamental work of Maxwell and Langmuir, and more recently in the fields of, a.o., drying, painting and patterning technologies, dehumidification, cooling technologies, desalination, and DNA synthesis.

## 2   Modeling the syringe water isotope delivery module

Here the dynamic behavior is modeled of the water vapor concentration (humidity) and isotope ratios of a low humidity-level generator (LHLG), such as the one described in the companion paper by Leroy-Dos Santos et al. (2020). Water isotope ratios are generally expressed in terms of the so-called "delta-value": $^x\delta_w := (^xR_w - ^xR_{VSMOW})/^xR_{VSMOW}$, the relative deviation of the abundance ratio of the rare isotope $x$ in reservoir $w$ with respect to the same ratio in the international stan-

dard material Vienna Standard Mean Ocean Water (VSMOW) (IAEA, 2017). In our case the relevant abundance ratios are $^2R_w = ([^2H]/[^1H])_w$ and $^{18}R_w = ([^{18}O]/[^{16}O])_w$. Although the model can just as well be applied to the $^{17}O$ isotope ratio, which is also measured by the laser spectrometer, these measurements were not considered here (in fact, $\delta^{17}O$ qualitatively tracks $\delta^{18}O$ very closely). Note that the isotope abundance ratios are exceedingly small numbers that can in practice be replaced by the corresponding molecular abundance ratios (Kerstel, 2004): $^{18}R_{VSMOW} = [^1H^{18}O^1H]/[^1H^{16}O^1H] \approx 2.005‰$ and $^2R_{VSMOW} = [^2H^{16}O^1H]/[^1H^{16}O^1H] \approx 0.3115‰$ (IAEA, 2006).

The LHLG instrument uses a commercial high-precision syringe pump system (Harvard 11 Pico Plus Elite) to push-in the plunger of a small-volume syringe. The needle of the syringe punctures a septum of a small evaporation chamber in which a steady air flow at a controlled pressure of 1 bar is maintained around the needle tip. Water being pushed through the syringe needle will start to form a droplet at the tip of the needle, provided the water flow is sufficiently high to overcome the evaporation from the exposed water surface inside the needle. Initially, as the water cap or droplet is still small, the evaporation rate from its surface into the surrounding dry air flow is smaller than the rate of water supply and the droplet continues to grow. As the droplet grows in size, its surface area increases and so will the rate of evaporation. Once steady state is reached, the evaporation of water from the surface of the droplet at the end of the syringe is exactly matched in quantity and isotopic composition by the supply of the standard water through the syringe needle.

Considering the isotopic composition of the evaporated water, it is clear that at the very beginning the isotopic composition of the meniscus (the droplet cap) equals that of the bulk water in the syringe. Also, in steady-state the isotopic composition of the vapor is identical to that in the syringe reservoir, for the simple reason of conservation of mass. In the transient regime, however, the isotopic fractionation occurring at the surface liquid-to-gas phase boundary implies an enrichment of the surface layer that first needs to diffuse inward. One thus expects to see a depleted vapor phase (relative to the reservoir liquid) as long as the droplet is growing. Inversely, if the water flow is reduced and the droplet shrinks, a temporary enrichment of the vapor is expected.

In order to model these dynamics quantitatively, and thus understand which factors control the magnitude of the transient signals, a pinned, sessile droplet is considered, with the shape of a partial sphere, as shown in Figure 1.

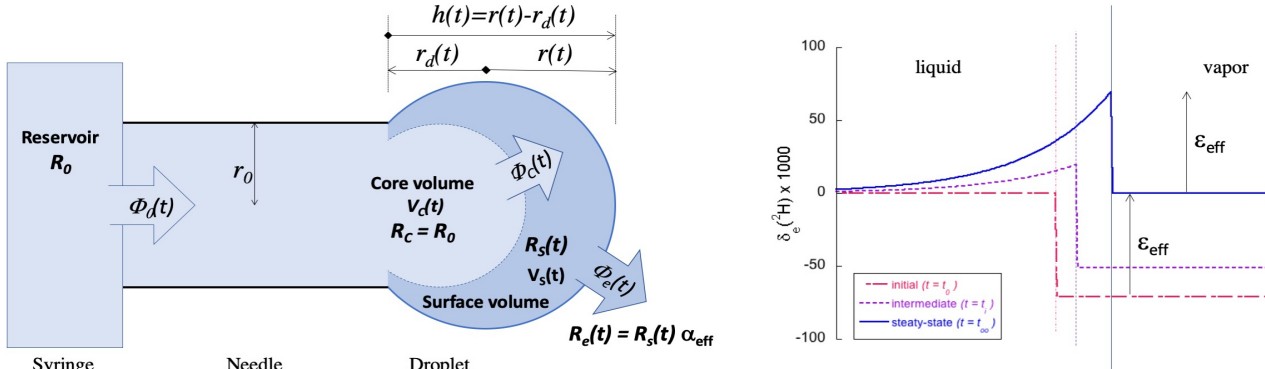

**Figure 1.** (a) Schematic representation of the ideal spherical droplet formed at the end of the syringe needle tip, illustrating the different reservoirs with volumes $V$, water fluxes $\Phi$, and isotope ratios $R$ involved in the model. Since $r^2 = r_0^2 + r_d^2$ and $r_d = r - h$, it follows that $r = \left(r_0^2 + h^2\right) / (2h)$. (b) The isotope ratio profile over the liquid to vapor boundary (left to right, with the thin vertical lines representing the growing water surface) at three instants in time if the water flux $\Phi_0$ from the syringe (with $\delta_0 = 0$) follows a step function with $\Phi_0(t) = 0$ for $t < t_0$, and $\Phi_0(t) = F > 0$ for $t > t_0$. The isotope fractionation is taken to be $\epsilon_{eff} \approx -71\text{‰}$ for $\delta^2\text{H}$. While the droplet is growing $\delta_e < \delta_0$. At $t = t_\infty$ the incoming water flux $\Phi_0$ equals the evaporated water flux $\Phi_e$ and $\delta_e = \delta_0$.

For completeness, it is assumed that only a fraction $f$ of the droplet volume (a boundary layer) becomes enriched. Later I will demonstrate that the best model results are obtained by assuming that the entire droplet becomes enriched ($f = 1$, see section 3.1), an observation that is further supported by considerations involving the relative speeds of isotopic diffusion and the water flow (section 4.1). Figure 1(b) shows the radial isotope concentration profile inside the droplet and the neighboring vapor following a step function in the flow rate from zero to some fixed value at $t = t_0$. The actual form of the profile is not important for the model and could just as well be approximated by a square profile. Four different bodies of water can thus be distinguished:

1. The syringe reservoir with a constant isotope ratio $R_0$ and an outgoing water flux equal to $\Phi_0(t)$ determined by the syringe pump speed,

2. The core volume of the droplet with an isotope ratio $R_c = R_0$ and a time-dependent volume $V_c(t)$. The water flux from the core to the surface layer of the droplet is given by $\Phi_c(t)$. Only in steady state $\Phi_c = \Phi_0$.

3. A fraction $f$ ($0 < f \leq 1$) of the total droplet volume $V_d(t)$ that will become enriched in the heavy isotope, $V_s(t)$, with isotope ratio $R_s(t)$:

$$V_s(t) = f \cdot V_d(t) \tag{1}$$

4. The evaporated water flux $\Phi_e(t)$ leaving the droplet with isotope ratio $R_e(t) = R_s(t) \cdot \alpha_{eff}$. Here the relevant isotope fractionation factor is that between the vapor and liquid phase water: $\alpha_{eff} = (1 + \epsilon_{eff}) < 1$.

A last essential part of the model is the assumption that the evaporation flux is proportional to the exposed surface area of the droplet:

$$\Phi_e(t) = k_e \cdot A_s(t) \tag{2}$$

Figure 1 gives a schematic representation of our model, indicating the relevant water volumes and inter-volume fluxes, as well as the isotope ratios $R$ of each volume. Solving the model *ab initio* is not difficult and will be shown to give a qualitatively and quantitatively satisfactory description of the dynamics under realistic conditions.

The free input parameters to the model are (a) the fraction $f$ of the droplet that becomes enriched, (b) the effective liquid-to-vapor fractionation factor $\alpha_{eff}$, and (c) the evaporation rate $k_e$. The initial estimates of these parameters were obtained from previous studies by Cappa et al. (2003) and Luz et al. (2009) for $\alpha_{eff}$, and Walton (2004) and Sefiane et al. (2009) for $k_e$. The values of these parameters that provide the best fit to the experimental data are subsequently rationalized in the Discussion, sections 4.1, 4.3, and 4.4, respectively.

The first task is to model the evaporated total water flux $\Phi_e(t)$ as a function of a variable input water flux $\Phi_0(t)$, driven by variations in the syringe pump speed. For this the mass balance equation for the non-compressible fluid is written out in discrete time with time step $dt$:

$$V_d(t + dt) = V_d(t) + (\Phi_0(t) - \Phi_e(t)) \cdot dt \tag{3}$$

The evaporation flux $\Phi_e(t)$ is a function of the droplet size through (2). For simplicity, the droplet at the tip of the needle is modeled as a partial sphere, a spherical cap. The surface area of the spherical-cap-shaped droplet is given by (see Figure 1):

$$A_s \equiv A_{cap} = 2\pi r h = \pi \left( r_0^2 + h^2 \right) \tag{4}$$

with the radius of curvature of the cap $r$, the cap height $h$ ($0 \leq h \leq 2r$), and the inner diameter $2r_0$ of the needle, all as defined in Figure 1. The volume of the droplet is equally a function of $h$:

$$V_d \equiv V_{cap} = \frac{\pi}{6} h \left( 3r_0^2 + h^2 \right) \tag{5}$$

$A_s(t)$, and thus $\Phi_e(t)$, can then be expressed in terms of $V_d$ by inversion of (5), with $h(V_d)$ being obtained as the only real root of the cubic equation, giving:

$$h(V_d) = \frac{\alpha^2 - 12u}{\alpha} \tag{6}$$

where

$$\alpha := \sqrt[3]{108v + 12\sqrt{12u^3 + 81v^2}} \tag{7}$$

and

$$u := 3r_0^2, \quad v := \frac{6V_d}{\pi} \tag{8}$$

The above already permits expressing both the droplet size (e.g., in terms of the droplet radius $r(t) = \left(r_0^2 + h(t)^2\right)/\left(2h(t)\right)$) and the evaporative water flux $\Phi_e(t)$ as a function of $V_d(t)$, and to subsequently calculate both as function of the time-dependent input water flux $\Phi_0(t)$ by numerical integration of Eq. (3).

Going one step further, a second mass balance equation is included in the model to account for the rare isotopologues (in this case either $^2\text{H}^{16}\text{O}^1\text{H}$ or $^1\text{H}^{18}\text{O}^1\text{H}$). First the rare isotope fluxes (identified by $^\star$) are expressed in terms of the total fluxes and the isotope ratio of the reservoir in question. For the three relevant fluxes (see Figure 1):

$$\Phi_0^\star = \Phi_0 \frac{R_0}{1 + R_0} = \Phi_0 \frac{R_{VSMOW}\left(1 + \delta_0\right)}{1 + R_{VSMOW}\left(1 + \delta_0\right)} \tag{9}$$

$$\Phi_c^\star = \Phi_c \frac{R_0}{1 + R_0} = \Phi_c \frac{R_{VSMOW}\left(1 + \delta_0\right)}{1 + R_{VSMOW}\left(1 + \delta_0\right)} \tag{10}$$

$$\Phi_e^\star = \Phi_e \frac{R_s \alpha_{eff}}{1 + R_s \alpha_{eff}} = \Phi_e \frac{R_{VSMOW}\left(1 + \delta_0\right)\alpha_{eff}}{1 + R_{VSMOW}\left(1 + \delta_0\right)\alpha_{eff}} \approx \Phi_e \frac{R_{VSMOW}\left(1 + \delta_0\right)\alpha_{eff}}{1 + R_{VSMOW}\left(1 + \delta_0\right)} \tag{11}$$

Recall that $R_w$ is the ratio of the abundance of the rare to the most abundant water isotope in the reservoir $w$ ($w = 0, c, s, e$ for respectively, the syringe and needle, the core of the droplet, droplet surface layer, and the evaporated water). Thus, the factors $(1 + R_w)$ in Eqs. (9), (10), and (11) account for the conversion from isotope abundance ratio to isotope concentration. Finally, the fractionation factor between the (evaporated) vapor phase water and the liquid $\alpha_{eff} \approx 1$ ($\epsilon_{eff} \ll 1$), making the approximation made in Eq. (11) a very good one.

Similar equations as (9), (10), and (11) hold for the different water reservoir volumes, allowing us to write for the volume of the isotopically enriched evaporating surface layer:

$$V_s(t + dt)\frac{R_s(t + dt)}{1 + R_s(t + dt)} = V_s(t)\frac{R_s(t)}{1 + R_s(t)} + \left(\Phi_c(t)\frac{R_0}{1 + R_0} - \Phi_e^\star(t)\right) dt \tag{12}$$

Substitution of:

$$\Phi_c(t) = \Phi_0(t) - \frac{dV_c(t)}{dt} = \Phi_0(t) - \frac{dV_d(t) - dV_s(t)}{dt} \tag{13}$$

And using the definition:

$$\psi(t) := \frac{R_s(t)}{1 + R_s(t)} \tag{14}$$

then yields:

$$\psi(t+dt) := \frac{1}{V_s(t+dt)} \left\{ V_s(t)\psi(t) + \Big( \big( V_s(t+dt) - V_s(t) \big) - \big( V_d(t+dt) - V_d(t) \big) \Big) \frac{R_0}{1+R_0} + \left( \Phi_0(t) \frac{R_0}{1+R_0} - \Phi_e(t)\psi(t)\alpha_{eff} \right) dt \right\} \quad (15)$$

where the approximation for $\Phi_e^\star(t)$ of Eq. (11) has been used.

The isotope ratio in the enriched fraction $f$ of the droplet volume (using Eq. (1) and an appropriate value of $f$) can now be calculated by integration of Eq. (15), while evaluating Eq. (1) to (4) at each time step. The isotope ratio of the evaporated water is then obtained as:

$$\delta_e(t) = \alpha_{eff} \left( 1 + \delta_s(t) \right) - 1 \tag{16}$$

with:

$$\delta_s(t) = \frac{R_s(t)}{R_{VSMOW}} - 1 \tag{17}$$

and:

$$R_s(t) = \frac{\psi(t)}{1+\psi(t)} \tag{18}$$

## 3  Results

The above model has been programmed in Mathcad (PTC Mathcad, 2020) and used to simulate data that were recorded with a high-precision, low-humidity water isotope spectrometer, named HiFi, described in Landsberg (2014) and Landsberg et al. (2014). Since we are specifically interested in the dynamic behavior of the water vapor source that feeds the spectrometer, it is necessary to take the response time of the spectrometer into account. This response is typically described by a double or even triple exponential. At humidity levels of several thousand ppmv (parts per million by volume) the initial (fast) response time of the bare spectrometer was determined to be in the range of 1 to 2 s for both the water concentration and the isotope ratios, with a second, slower exponential response of the order of 15 s. However, in the configuration of this study, and at the lower water concentrations of a few hundred ppmv, the response time is significantly longer, especially for the $\delta^2\mathrm{H}$ isotope ratio. These response times were measured using the previously mentioned prototype humidity source (see section 1), a predecessor of the isotopic humidity generator described in Leroy-Dos Santos et al. (2020), which was, however, equipped with two independent syringe pumps, enabling rapid switching between two different water sources using a 2-position, 4-port valve (Valco EUDA-4UWE) just before the spectrometer (Landsberg, 2014). The humidified air stream was sent either to the spectrometer or to a waste pump. The isotope response was determined by switching between two very different water standards, assuring a high signal-to-noise-ratio of the measurements, while keeping the concentration constant at about 600

175 ppmv. The standard waters used were working standards of the Groningen Center for Isotope Research (CIO), known as GS-48 ($\delta^{18}O= -6.3\permil$, $\delta^2H= -43\permil$) and BEW-2 ($\delta^{18}O= 795\permil$, $\delta^2H= 5983\permil$). It is noted that despite careful storage these absolute isotopic compositions can no longer be guaranteed with the precision specified by the CIO, as the standards were used previously for other experiments. For all measurements shown here, the water isotope analyzer was calibrated with respect to the same water (GS-48) used for the evaporation measurements, resulting in relative isotope deviations ($\delta$-values) equal to

180 zero in steady-state conditions. The absolute isotope ratios are therefore not relevant. In any case, the drift of the standards was estimated to be less than $1\permil$ for $\delta^2H$ and less than $0.2\permil$ for $\delta^{18}O$ (due to possible Rayleigh distillation).

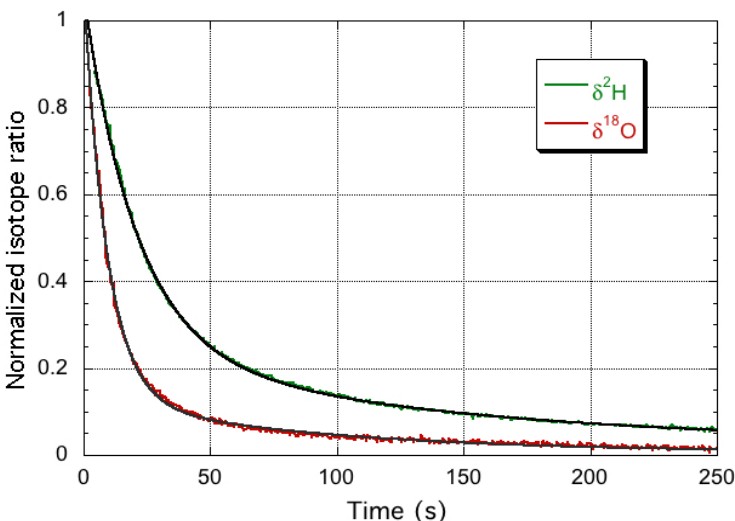

**Figure 2.** The normalized response curves of the spectrometer for switching between the GS-48 and BEW-2 isotope standards, both prepared as a mixture of $\sim$600 ppmv water vapor in dry air. The experimental data is fit with a double exponential yielding for the fast decay times 9.2 s and 20.7 s for $\delta^{18}O$ (red curve) and $\delta^2H$ (green curve), respectively. It is noted that the update frequency of the water isotope spectrometer is 2 Hz (Landsberg et al., 2014).

The instrument isotope response curves are shown in Figure 2, while the double exponential fit parameters are summarized in Table 1. Whereas the total water vapor concentration and $\delta^{18}O$ show practically the same time response, $\delta^2H$ is about twice as slow, due to different time constants for the surface adsorption processes. Although at much higher humidity, Steen-Larsen

et al. (2014) observed a qualitatively similar behavior. In the following sections, the time response of the spectrometer is taken into account, by convolution of the simulated response of the humidity generator with the calculated impulse response of the spectrometer that corresponds to the step response of Figure 2, before comparison to the corresponding experimental data.

**Table 1.** Parameters of the double exponential fit to the measured instrument response for $\delta^{18}O$ and $\delta^2H$. The water vapor concentration is observed to closely follow the $\delta^{18}O$ behavior and was modeled with the $\delta^{18}O$ parameters.

|  | $\tau_1$ (s) | $A_1$ | $\tau_2$ (s) | $A_2$ |
|---|---|---|---|---|
| $\delta^{18}O$, [H$_2$O] | 9.2 | 0.88 | 104 | 0.12 |
| $\delta^2H$ | 21 | 0.80 | 145 | 0.20 |

## 3.1 Humidity and isotope step responses

The model detailed in section 2 was first used to simulate the dynamic behavior of the combination of the LHLG and the HiFI isotope analyzer, while the LHLG was programmed to generate small humidity steps of about 200 ppmv around an absolute value of roughly 400 ppmv. The simulated water vapor concentration response was fit to the experimental data by adjusting the evaporation rate $k_e$, the only free parameter in this case (see the top panel of Fig. 3). I will discuss the rationale for the values of $k_e$ determined in this study later in section 4.4. Having fixed the evaporation rate at an optimal value of $k_e = 3 \ \mu$m/s, the next step is to confirm that the isotope responses are modeled correctly, taking into account that both the $\delta^2H$ and $\delta^{18}O$ simulated responses also depend on the fraction $f$ of the droplet volume that becomes enriched, as well as the effective liquid-to-vapor fractionation factor $\alpha_{eff}$. Since it can be expected that the entire droplet becomes enriched in the heavy isotopologues, the logical starting point is the assumption that $f = 1$. As we will see shortly, this choice is validated by the experimental observations. It will subsequently be rationalized by theoretical considerations in section 4.1.

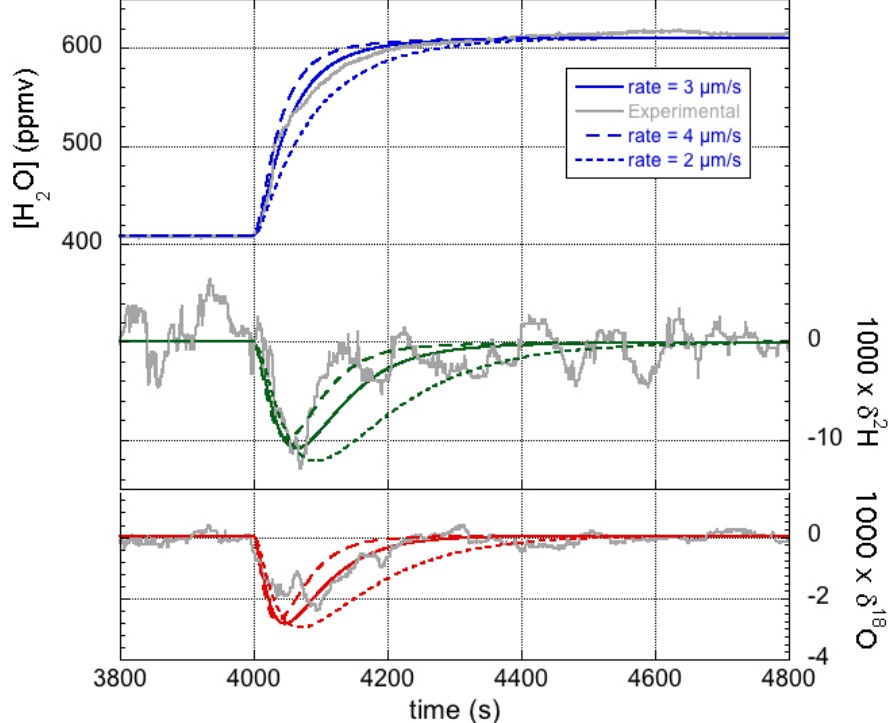

**Figure 3.** Experimental data (gray curves) and model simulations of the humidity (blue traces, top panel) and isotope response curves (green for $\delta^2$H and red for $\delta^{18}$O, lower panel) for three different values of the evaporation rate $k_e$. The best fit is obtained for $k_e \approx 3 \ \mu$m/s, whereas a higher (lower) value results in a simulated dynamic response that is too fast (slow) compared to the measured response.

As to the fractionation factors, at the very low relative humidity of the experiment ($h \approx 0.01$), the effective fractionation factors $\alpha_{eff}$ can be written as the product of a diffusion fractionation factor $\alpha_{diff}$ and an equilibrium fractionation factor $\alpha_{eq}$ (Cappa et al., 2003). Moreover, the diffusion fractionation factor can be related to the ratio of the molecular diffusivities (Stewart, 1975), such that one may write:

$$^x\alpha_{eff} = \ ^x\alpha_{eq} \left( \frac{D(x)}{D(a)} \right)^n \tag{19}$$

As before the label $x$ refers to the rare isotope or isotopologue ($^2$H and $^{18}$O or $^2$H$^{16}$O$^1$H and $^1$H$_2^{18}$O) and $a$ to the abundant isotope or isotopologue ($^1$H and $^{16}$O or $^1$H$_2^{16}$O). The effective fractionation factors for $^2$H$^{16}$O$^1$H and $^1$H$_2^{18}$O are thus not independent, but are determined by the single parameter $n$. The exponent $n$ in (19) equals unity in the case of laminar flow, and zero in the case of fully turbulent flow. The equilibrium fractionation factors were accurately determined by Horita and Wesolowski (1994), and their values at 35 °Care used here, the temperature of the evaporation chamber. The diffusivities

**Table 2.** Effective fractionation factors as a function of the flow parameter $n$. For $n = 0$ the fractionation factors are equal to the equilibrium values at 35 °C, such as they were determined by Horita and Wesolowski (1994).

| $n$ | 0 | 0.43 | 1 |
|---|---|---|---|
| | turbulent | intermediate | laminar |
| $^2\alpha_{eff}$ | 0.9370 | 0.9288 | 0.9181 |
| $^{18}\alpha_{eff}$ | 0.9915 | 0.9800 | 0.9650 |

were determined by Cappa et al. (2003) and more recently by Luz et al. (2009). The more recent values are used here, but
the difference is minimal for our purpose (Cappa et al. (2003) predict only slightly lower values of $\alpha_{eff}$ in the laminar limit
of $n = 1$). Table 2 gives the values of the effective liquid-to-vapor fractionation factors for three different values of the flow
parameter $n$ and Fig. 4 shows the corresponding model simulations compared to experimental data. The $\delta^{18}O$ simulation shows
a larger effect of changing $n$ than the $\delta^2H$ simulation. In contrast, changing the values of $f$ has the same relative effect on both
simulations (not shown in 4). With $n = 0.43$ and $f = 1$ a good fit to both isotope response curves is obtained. One may thus also
conclude that the data support the theoretical finding (section 4.1) that $f = 1$. A good observer will have noted the difference
in magnitude and noise level of the $\delta^2H$ and $\delta^{18}O$ responses shown in Figs. 3 and 4. Both are easily explained by noting that
the isotopic fractionation is larger for $^2H$ than for $^{18}O$, while at the same time the signal-to-noise ratio of the deuterium feature
detected by the infrared spectrometer is significantly lower than that of $^1H_2^{16}O$, which in turn is directly related to the lower
abundance of $^2H^{16}O^1H$ with respect to $^1H^{18}O^1H$ in the natural water sample.

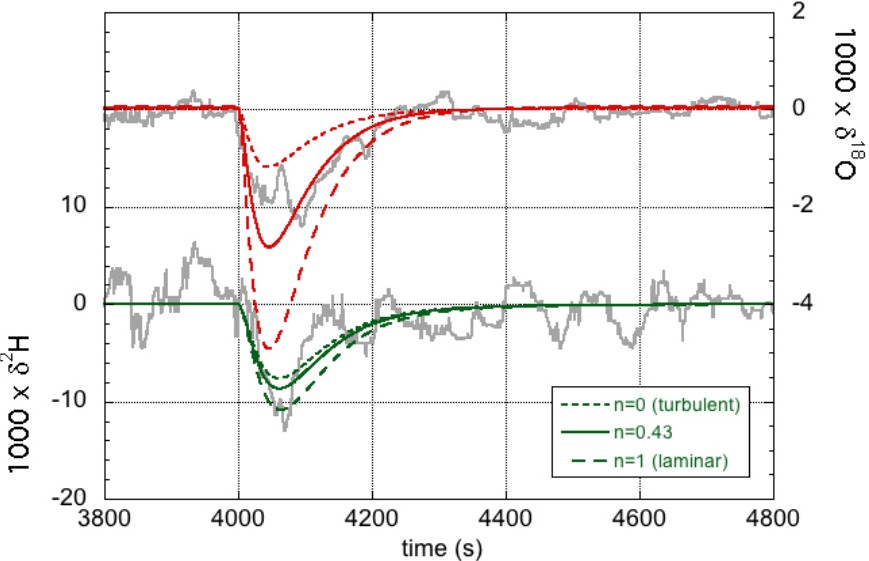

**Figure 4.** Experimental data (gray curves) and model simulations of the isotope response curves (green for $\delta^2H$ and red for $\delta^{18}O$) for three different values of the flow parameter $n$. $n = 0$ (dotted lines) corresponds to the turbulent flow limit, while $n = 1$ (dashed lines) corresponds to the limit of laminar flow. For $n = 0.43$ (solid lines) a good fit is obtained for both isotopes.

## 3.2 Dynamic response under non-ideal conditions

The LHLG prototype was modified immediately following the experiments presented in the previous section. Notably, it was deemed that the bore of the aluminum injector chamber that accepts the syringe's needle was too narrow. With an internal diameter of only 2 mm, careful guiding of the needle, and consequently a precise positioning of the syringe, was needed to avoid occasional contact of the droplet with the chamber wall. This also limited the maximum droplet size, and therewith the volume mixing ratio (humidity level) that could be attained to roughly 1000 ppmv. The injection chamber was therefore replaced by a stainless steel sample cylinder with volume 75 mL and Sulfinert hydrophobic coating (Restek 304L-HDF4-75). Because the flow velocity is now significantly lower, the coating serves to minimize the memory effect due to surface adsorption of water molecules. In addition, a section of PTFE tubing was added between the syringe (Hamilton 84853) and the removable needle to make the alignment better manageable. This gave initially rise to unexpected results that were attributed to the appearance of small air bubbles in the water injection line. These problems were later resolved by re-engineering the LHLG as described in Leroy-Dos Santos et al. (2020). These "useless" results that otherwise might have been discarded are reported here anyways because they nicely demonstrate the ability of the model to simulate the behavior of this non-ideal instrument, and thus validate the model under a different operating regime.

During similar experiments as reported in the previous section 3.1, recording the response of the LHLG following small steps in the flow of injected water, relatively large sinusoidal oscillations were observed with a period that matched the revolution speed of the lead screw of the precision pump. I submit that these oscillations become prominently visible when small imperfections of the lead screw combine with small air bubbles present in the water injection line, possibly amplified by viscous resistance of the liquid inside the water line and needle. Whatever the precise underlying mechanics, a sinusoidal variation of the water flow was modeled with a period equal to one revolution of the screw drive. The amplitude and phase of the (possibly amplified) lead screw imperfection was chosen to yield a simulation that best matched the observed amplitude of the oscillations. The only other parameter that needed adjustment was the evaporation rate. A value of $k_e = 1$ $\mu$m/s was found to produce a simulation that best matched the water vapor concentration response when the pump was switched between different water flow rates, as seen in the upper panel of Fig. 5. The lower evaporation is due to the lower flow velocity of the air around the droplet (see section 4.4).

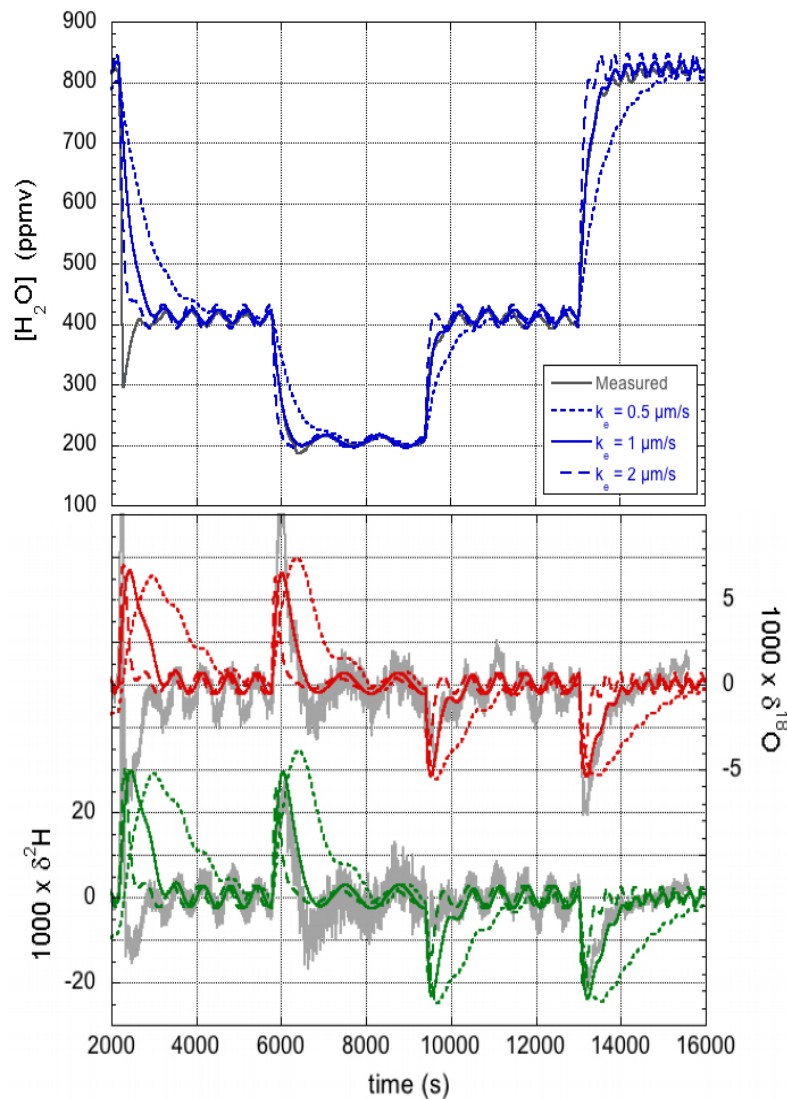

**Figure 5.** Humidity (upper panel) and isotope responses (lower panel) of the modified LHLG subject to stepwise changes in the water flow rate. The best fit (blue for the humidity response, green for $\delta^2$H and red for $\delta^{18}$O) to the experimental data (gray curves) is obtained for an evaporation rate $k_e = 1$ $\mu$m/s. The overshoot in the first measured downward humidity transition and the corresponding inverted isotope response are most likely due to an air bubble in the water line.

The corresponding response of the isotope ratios is shown in the lower panel of Fig. 5. It may be clear that the correspondence between simulation and experiment is (already) satisfactory, considering that no further parameter adjustments were made. The simulation will be further refined in the Discussion, section 4.3.

## 4    Discussion

### 4.1    Droplet isotopic enrichment

Here support is provided for the observation of an enrichment in the heavy isotopologues of the entire droplet, and not just in a surface layer of limited thickness. Referring to Fig. 4 (for which $n = 0.43$, i.e., $^{18}\alpha_{eff} = 0.98$, and $f = 1$), in principle the same amplitude of the modeled response can be obtained by assuming fully laminar flow ($n = 1$), ánd assuming that a much smaller fraction of the droplet becomes enriched in the heavy isotopes. This gives, however, a less satisfactory fit to the data, as shown in Fig. 6. Notably, the response simulated with $n = 1$ (i.e., $^{18}\alpha_{eff} = 0.9650$) and $f = 0.5$, reached the same maximum

amplitude, but is clearly narrower than the experimental curve. Importantly, this is also not what is predicted based on the speed of isotopic diffusion in the droplet.

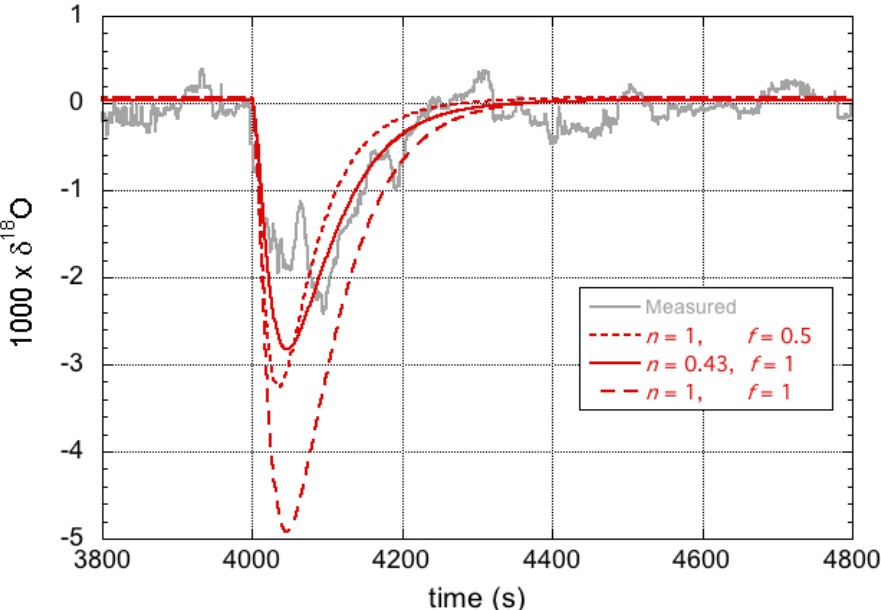

**Figure 6.** Experimental data and model simulations of the $^{18}O$ response curves for two different values of $\alpha_{eff}$ and two different values of $f$.

To see that in present experiment $f$ should be equal to unity I consider that the enrichment occurring at the surface of the droplet will diffuse inwards, resulting in an isotope gradient inside the droplet with a characteristic diffusion length given by Bird et al. (2006):

$$L = 2\sqrt{D \cdot t} \tag{20}$$

with $D$ the diffusion coefficient and $t$ time. Differentiation of (20) yields the velocity of the diffusion front:

$$v_{diff} = \sqrt{\frac{D}{t}} \tag{21}$$

The diffusion coefficients of HDO and H$^{18}$OH in water have been measured by Horita and Cole to be 2.34 $10^{-9}$ m$^2$/s and 2.66 $10^{-9}$ m$^2$/s, respectively (Horita and Cole (2004)). This shows that diffusion over lengths comparable to the size of a typical droplet (0.1 mm) takes place on a time scale of the order of 1 s. It is therefore likely that the entire droplet becomes isotopically enriched, rather than just a surface layer: $f = 1$.

## 4.2 Back Diffusion

The question arises whether the diffusion is strong enough to allow the isotopic enrichment to propagate all the way to the syringe reservoir. To answer this question the diffusion velocity of (21) is compared to the flow velocity inside the syringe needle:

$$v_{flow} = \frac{\Phi_0}{A_0} \tag{22}$$

with $\Phi_0$ the water flux through the syringe needle and $A_0$ the needle's internal cross sectional area. After a characteristic time $t_e$, the diffusion velocity will have become smaller than the flow velocity, at which point in time the diffusion front does not further penetrate into the needle. This characteristic time equals:

$$t_e = D \left(\frac{A_0}{\Phi_0}\right)^2 \tag{23}$$

With typical values for the prototype instrument (an inner diameter of 464 μm for the gauge 26 needle and a low water flux of about 100 nL/min; nL = nanoliter), the flow velocity inside the needle is about 0.6 mm/min, such that $t_e \approx 25$ s. Eq. (20) then shows that the enrichment propagates about 0.5 mm into the 51-mm long needle. Moreover, at the given flow rate, it takes about 600 s to arrive at the typical droplet size of 10 μL. In this case, the isotopic diffusion into the needle thus stops already before steady-state is reached. Even at the lowest water flow rates of about 0.1 nL/min, the diffusion can be stopped well within the length of the needle (if necessary by reducing the needle inner diameter). It is thus unlikely that the isotopic composition of the syringe reservoir would change due to back-diffusion of heavier isotopologues. This was also confirmed experimentally by bringing the same liquid standard material into the vapor phase with both the LHLG and a commercial humidity generator (Picarro SDM) at time intervals of one month and not observing any difference between the measurements (within the measurement precision of 0.2‰ and 1‰ for $\delta^{18}$O and $\delta^2$H, respectively) (Leroy-Dos Santos et al., 2020).

### 4.3 Fractionation factors

The effect of the precise values of the $^2$H$^{16}$O$^1$H- and H$_2^{18}$O-isotopologue effective fractionation factors on the simulations was already discussed to some extent in section 3.1, where it was found that the best match with experiment is obtained by assuming fractionation factors that correspond to an intermediate case between laminar and turbulent flow (characterized by $n = 0.43$). This can be rationalized by estimating the Reynolds number for the flow around the water droplet, $R_e = \rho v L / \mu$. In the previous formula $\rho \approx 1.25$ kg m$^{-3}$ is the density of the air flowing around the needle and droplet; $v = 1.6$ m/s is the velocity of the air around the droplet inside the narrow-bore chamber (inner diameter 2 mm), given the air flow of 300 mL/min (STP); $L \approx 0.5$ mm is the diameter of the droplet; and $\mu = 18.3$ $\mu$Pa·s is the dynamic viscosity of air at 35 °C. With these values $R_e \approx 60$. This contrasts with a value of $v \approx 0.007$ m/s and $R_e \approx 0.2$ for the case of the about 30-mm internal diameter steel cylinder used in the modified instrument. The latter case should thus be much closer to the limit of fully laminar flow. The simulations of Fig. 5 were therefore repeated, but now with the fractionation factors for $n = 1$ (see Table 2). The new simulations are shown in Fig. 7.

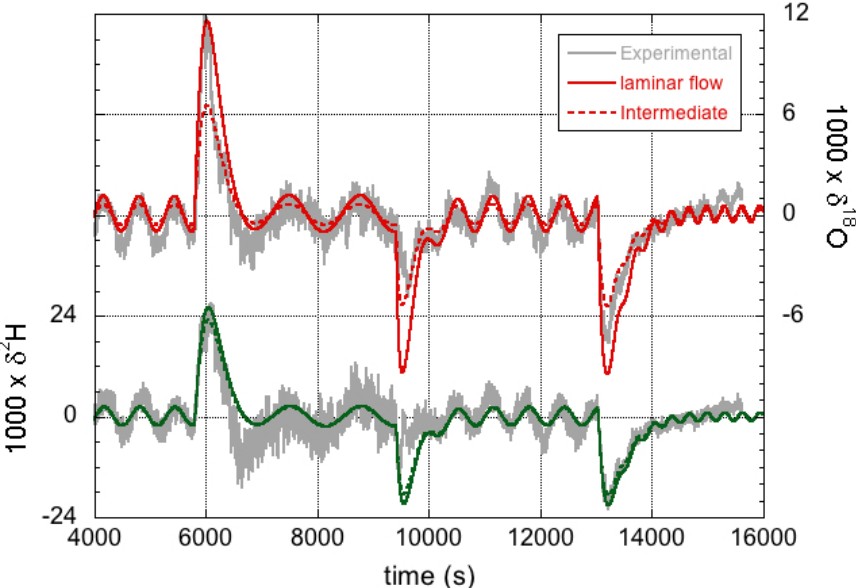

**Figure 7.** The isotope response of the modified LHLG subject to stepwise changes in the water flow rate. Improved simulations, compared to those of Fig. 5, are obtained with $k_e = 1$ $\mu$m/s and effective fractionation factors for the limiting case of fully laminar flow.

Whereas the differences for $^2$H are minor, the effect of the larger $^{18}$O-fractionation (i.e., the smaller liquid-to-vapor fractionation factor, which is smaller than unity) in the laminar flow regime is clearly visible, and arguably provides a slightly better fit to the data, primarily during the water vapor concentration changes, as can be seen in Fig. 7. It should be noted, however, that

in the regions of oscillatory behavior in between the concentration steps, the fit could also have been nudged by adjusting the amplitude of the lead screw modulation. Still, the results of section 3.2 are just as well, and most likely better, described (than shown in Fig. 5) by assuming fully laminar flow.

### 4.4 Evaporation rate

The two experiments discussed here in sections 3.1 and 3.2 required rather different evaporation rates to simulate the data with our model, $k_e \approx 3$ $\mu$m/s and 1 $\mu$m/s, respectively. The difference is clearly related to the different Reynolds numbers or, more directly, the different dry air flow velocities of 1.6 m/s and 0.007 m/s. In fact, the values are in reasonable agreement with the results reported by Walton (2004). Although his measurements were recorded at only a small number of air temperatures and flow velocities, values applicable to our situation can be estimated by linear extrapolation of the observed rates as a function

of flow velocity, and fitting a (weakly) quadratic dependence on the temperature to the data collected at a fixed flow velocity of 1 m/s. In Fig. 8 selected data of Walton are presented together with the estimated values for our case. One thus predicts a rate of 5.2 $\mu$m/s at a flow velocity $v = 1.6$ m/s, and of 1.3 $\mu$m/s at $v = 0$ m/s, higher than the experimental values found here. So far it has been assumed that the droplet is at the same temperature as the evaporation chamber, but it cannot be excluded that the actual droplet temperature is lower, especially in the high velocity case. However, the study by Sefiane et al. (2009)

measured an evaporation rate at 22 °C and 1 bar corresponding to $\sim 4$ $\mu$m/s, very close to the value extrapolated from the data of Walton (2004) at 25 °C. It is noted that the observation of a slightly lower evaporation rate than found by Walton (2004), and also Sefiane et al. (2009), is in agreement with the experimental sessile droplet, as it sits on the beveled tip of the syringe needle (Landsberg, 2014; Leroy-Dos Santos et al., 2020), having a somewhat smaller surface to volume ratio than the one that is modeled. It should also be mentioned that it is unlikely that the difference with the observations by Walton (2004) or Sefiane

et al. (2009) are due to an under-estimation of the spectrometer humidity response time, as it is difficult to imagine a response time of the water vapor concentration that is slower than that of the isotope ratios.

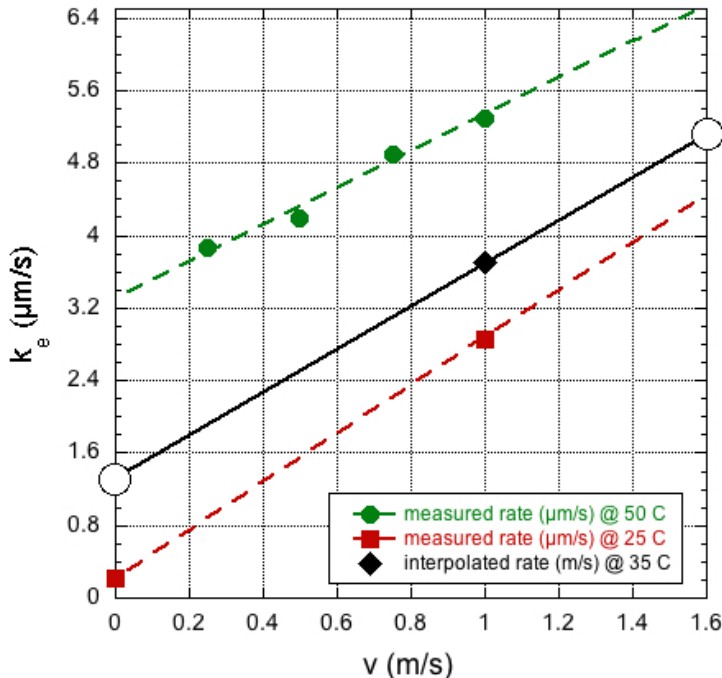

**Figure 8.** Evaporation rate measurements made by Walton (2004) as a function of flow velocity at 25 °C (red solid squares) and at 50 °C (green solid circles) and one interpolated point at 35 °C (black solid diamond), leading to the extrapolated estimates for our experiment (open circles).

## 5  Conclusions

I have shown that the dynamic behavior of a humidity generator based on droplet evaporation can be accurately modeled. Confrontation with experimental data of the water vapor concentration and two isotopic ratios as a function of the injected water flow, enables the determination of physically realistic values of the droplet evaporation rate and the liquid-to-vapor isotope fractionation factors. However, the signal-to-noise ratio of the water isotope analyzer at the very low humidity levels investigated is not quite sufficient to make very precise determinations of the fractionation factors. But recent developments in ultra-precise and ultra-sensitive isotope measurements (e.g., Stoltmann, 2017; Kassi et al., 2018) will enable to deliver more precise values by at least an order of magnitude. What may appear as a bit of a quixotic study of evaporating water droplets, may thus in fact permit measuring not only the evaporation rate, but also the effective fractionation factors, and therewith also isotopologue dependent diffusivity ratios, in the evaporation of small sessile droplets. Apart from this potentially

new application, it is highly satisfactory to be able to accurately simulate the dynamic behavior of the LHLG with few free parameters, and under rather different operating conditions.

*Code availability.* Please contact the author if you wish to obtain a copy of the Mathcad code.

*Author contributions.* EK developed the model and Mathcad code, and applied these to the data obtained in Grenoble using instrumentation built by Janek Landsberg, Daniele Romanini and EK.

*Competing interests.* The author declares to not have any competing interests.

*Acknowledgements.* I am indebted to past and present colleagues and students. In particular, Janek Landsberg was instrumental in building the prototype humidity generator, with valuable contributions coming from Daniele Romanini and Samir Kassi. Janek Landsberg and Marine
Favier recorded some of the experimental data shown here. Amaelle Landais, Andreas Zahn, and David Walton provided valuable feedback on the manuscript, as did an anonymous reviewer.

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
