# Peer review of "Modeling the dynamic behavior of a droplet evaporation device for the delivery of isotopically calibrated low-humidity water vapor"

_Atmospheric Measurement Techniques, 2020_

## Referee Comment (RC1) · David Walton (Referee) · 4 Jan 2021

The Author reports on an important aspect of instrumental analysis namely calibration, in that the reliability of all analytical data is only as good as the standard it's compared against. A vapour calibration device for isotope analysis and the associated mathematical model are presented by the Author. To generalise, although the positive aspects of the calibration device and model are dealt with, it would be beneficial if the Author further addresses some of the system's limitations. For example, what tolerance or accuracy does the device operate under? Can the Author quote a reliability of plus or minus X%? How applicable is the calibration in the low temperature Artic conditions

reported, when the model employs diffusional data derived at 35oC? What part of the calibration curve does the device operate under, is it similar to real-word conditions? If manufactured commercially, would variance in reproducibility and accuracy between manufactured devices be a problem? Would such devices need to be re-calibrated and how often? The Author may also like to consider the following... • Line 1: Consider using the term "water vapour" rather than "water concentration". • Line 2: "isotope ratios"... What isotope ratios are being discussed, oxygen, hydrogen... not actually mentioned until line 91. • Line 3: "nL-droplet" ... Define the term nL, is it nano-litre? • Line 8: Consider replacing "We" with... "it can be shown". Also, there's a tendency for the Author to use the plural "We" throughout the whole narrative, rather than the singular. • Line 23: Consider revising the English, remove... "we provide" and replace with "a theoretical quantitative model is presented". • Line 24: Title spelling - also see main title spelling: Modelling has two l's, and consider removing "syringe water" and replacing with... "Modelling the syringe water isotope delivery module" • Line 25: Consider re-wording the statement and check spelling of behaviour, for example... "The dynamic behaviour of the water concentration (humidity) and isotope ratios of a low humidity-level generator (LHLG), such as described in the companion paper by Leroy-Dos Santos et al. (2020), are modelled here." • Line 34: "standard water" is an ill-defined term, does the Author mean calibration standard? • Line 38 and 219 to 220: Can the Author put forward a mechanism that drives the isotope fractionation? Is it diffusional in nature, and would surface molecular diffusion play any part in this mechanism i.e. where Nu and Sh approaches a value of 2. • Lines 35 to 40: How does the Author know the isotope composition at any one time during the instrument's start-up, steady-state equilibrium, and variations in steady-state operation e.g. due to changes in droplet size. How was this analysed? What is the frequency or need for isotope re-calibration? • Line 43: Does the needle tip have a bevelled tip as most Hamilton syringes do, as this would affect the shape, mass and consequently the transport properties of the suspended droplet. Was the needle tip profile engineered in any way? • Line 60 and 70: To what

extent has the Author considered the model's response in terms of heat and mass transfer, to the following. . . o Changes in droplet temperature. o Molecular diffusion, convection, and conduction via the needle to and from the droplet. o Surface tension and surface energy. . . at 0.1mm in diameter these energies must be high. o Is there any isotopic absorption-desorption equilibrium at the evaporation chamber wall? o Are there any static effects that need to be considered? o The droplets are hemispherical in geometry; would a spherical droplet experience enhanced heat and mass transfer due to droplet instability such as oscillations or distortions in the chamber airflow? o What type of water did the Author use. . . distilled, degassed, isotope enriched? o How is the evaporation chamber temperature is maintained? What are the tolerances? • Line 124: The following statement raises questions about the reliability of the data produced by the instrument. . . "a home-built, low-humidity water isotope spectrometer" • Line 136: Similarly, the following statement raises questions of reliability in the calibration process e.g. do these standards deteriorate with time? . . . "The standard waters used were left-over working standards" • Line 160 and 167: Table 2, data evaluated at 35oC. Is it valid for the Author to use this data? What is the modelling temperature and operational temperature of the device? • Line 186: The Author uses the term "useless results". . . if used to validate data, whether in a positive or negative sense, the data can hardly be described as useless!

Please also note the supplement to this comment:
https://amt.copernicus.org/preprints/amt-2020-428/amt-2020-428-RC1-supplement.pdf

---

## Referee Comment (RC2) · Anonymous Referee #2 · 14 Feb 2021

The author presents a mathematical model to calculate the water isotope ratio which has been measured with a new developed calibration instrument described recently in another work published by Leroy-Dos Santos et al. (2020) in AMTD. The results of the presented simulations fit quite well with the measurements of that particular calibration device. I think it would have been more suitable to publish this work as an add-on to the published paper mentioned above. I don't see the unique selling point of this work and the added value of it for the community. Therefore I suggest a major revision. In any case I have several comments the author should consider to incorporate into the manuscript before publication. Major issues: 1) For a better understanding the author should briefly describe in the introduction why the measurement of the water

isotope is important and what insights you gain from it. 2) It would be good to describe the meaning of $\delta$ for the readers not so familiar with the subject 3) The derivation of the formulas are not always easy to follow, please prepare a better description. 4) Generally please replace 'water concentration' with 'water vapour concentration'. 5) All figures should be enlarged. The legend of the figures are too small. 6) The colours of the graphs changes from one figure to the next. The author should consider to use the same colour for the simulation in all figures. 7) Figure 4 why does $\delta$2H oscillate so much more than $\delta$18O. Please give an explanation.

Minor issues: 1) line 66-69 the use of numbers in brackets could lead to the misunderstanding that the author refers to the number of the appropriate formulas, consider using other annotations. 2) line 136 The standard waters used were left-over? Consider a better formulation.

---

## Author Comment (AC1) · 16 Mar 2021

Reply to the Interactive comment by David E. Walton (Referee 1) on the manuscript "Modeling the dynamic behavior of a droplet evaporation device for the delivery of isotopically calibrated low-humidity water vapor" (https://doi.org/10.5194/amt-2020-428).

I would like to start with thanking the reviewer for his thorough review of the manuscript and the many helpful comments. As outlined here below, these will be implemented in the final manuscript where applicable. Here below I will copy the text of the review, then provide my reply immediately following each item of the review in blue text (see

supplement).

The Author reports on an important aspect of instrumental analysis namely calibration, in that the reliability of all analytical data is only as good as the standard it's compared against. A vapour calibration device for isotope analysis and the associated mathematical model are presented by the Author. To generalise, although the positive aspects of the calibration device and model are dealt with, it would be beneficial if the Author further addresses some of the system's limitations. For example, what tolerance or accuracy does the device operate under? Can the Author quote a reliability of plus or minus X%? How applicable is the calibration in the low temperature Artic conditions reported, when the model employs diffusional data derived at 35oC? What part of the calibration curve does the device operate under, is it similar to real-word conditions? If manufactured commercially, would variance in reproducibility and accuracy between manufactured devices be a problem? Would such devices need to be re-calibrated and how often?

Addressing the above aspects of the prototype device as suggested by the reviewer is in fact done in the companion, experimental paper by Leroy Dos Santos et al. that describes the improved version of the device used to produce the data for this paper (Leroy-Dos Santos, C., Casado, M., Prie, F., Jossoud, O., Kerstel, E., Kassi, S., Fourre, E., and Landais, A.: A dedicated robust instrument for water vapor generation at low humidity for use with a laser water isotope analyzer in cold and dry polar regions., Atmos. Meas. Tech. Discuss., in review, https://doi.org/10.5194/amt-2020-345, 2020.). The revised and recently accepted (but not yet on-line) version of the Leroy-Dos Santos paper discusses the tolerances of, for example, the injection chamber temperature and pressure on the water vapor concentration (= humidity level) produced by the device, and the range over which the device can be used. It also discusses the long-term stability of the device in terms of an Allan Variance analysis, and shows that there is no observable isotope ratio drift over the duration of its employment in Antarctica (From the Leroy-Dos Santos paper: "The water in the water reservoirs is sampled every month to check its isotopic composition and renewed when the level of water is below half the maximum level. A maximum evolution of the isotopic composition of the lab-standard filling the water reservoirs has been observed as 0.05‰ and 0.5‰ respectively for d18O and dD over a 2-month period"). The calibration of the isotope scale can be as good as the precision with which the isotope ratios of the water standard are known (normally determined by repeated mass spectrometer comparison to international standards), combined with the precision afforded by the optical spectrometer (under the working conditions in the field). Long-term stability requires only that the bulk water reservoir is hermetically sealed. As mentioned, we did not observe derivation of the isotopic composition of the bulk water over periods as long as 1 year [Leroy2020]. It is important to note that under steady-state operation the isotope ratios of the evaporated water in our device are identical to those of the bulk water reservoir. Therefore one does not expect to see any differences in this respect between different copies of the device. Normally, these devices will serve foremost to calibrate the isotope ratio scale of the optical spectrometers to which they are interfaced; the humidity level is only of secondary concern. Still, the companion paper shows that one can expect an accuracy of the humidity level of about 2%; the error being determined by the air mass flow measurement/control (specified to be better than 1%), and knowledge of the syringe injector speed (i.e., the liquid water flow towards the injection chamber, specified precision of 0.5%).

The Author may also like to consider the following. . . Line 1: Consider using the term "water vapour" rather than "water concentration".

I agree that the use of the term "water concentration" is not appropriate without the implicit or explicit addition "in the air (stream)". I propose to use the terms "water vapor concentration" (as proposed by Reviewer #2) or "humidity" there where "in the air" is not specified.

Line 2: "isotope ratios"... What isotope ratios are being discussed, oxygen, hydrogen... not actually mentioned until line 91.

Thank you for pointing this out. I will specify which isotope ratios are considered much earlier (the Introduction) in the final manuscript.

Line 3: "nL-droplet" . . . Define the term nL, is it nano-litre?

Yes, nL is used here as an abbreviation for nanoliter. I will add the full term at the first appearance in the text.

Line 8: Consider replacing "We" with. . . "it can be shown". Also, there's a tendency for the Author to use the plural "We" throughout the whole narrative, rather than the singular.

The use of "we" was thought to be more polite than the singular form, which fails if it is rather perceived as a form of false modesty. I will thus be glad to follow the suggestion of a native English speaker to refrain from nosism and instead reword the text in passive form, using the impersonal "one", or the singular "I".

Line 23: Consider revising the English, remove. . . "we provide" and replace with "a theoretical quantitative model is presented".

Thank you for this suggestion, to be implemented in the final version.

Line 24: Title spelling - also see main title spelling: Modelling has two l's, and consider removing "syringe water" and replacing with... "Modelling the syringe water isotope delivery module"

I have tried to use the US spelling consistently throughout the text. I will leave the decision as to whether to follow US or UK spelling up to the Editor.

Line 25: Consider re-wording the statement and check spelling of behaviour, for example... "The dynamic behaviour of the water concentration (humidity) and isotope ratios of a low humidity-level generator (LHLG), such as described in the companion paper by Leroy-Dos Santos et al. (2020), are modelled here."

Thank you for this suggestion. Will be adopted, with the exception perhaps of the US

vs. UK spelling of behavior(u)r.

Line 34: "standard water" is an ill-defined term, does the Author mean calibration standard?

The term "water standard" is a commonly used in the isotope community to denote a calibration material, in this case water. In line 34, there is actually no need for the "standard" specifier. I propose to delete is altogether.

Line 38 and 219 to 220: Can the Author put forward a mechanism that drives the isotope fractionation? Is it diffusional in nature, and would surface molecular diffusion play any part in this mechanism i.e. where Nu and Sh approaches a value of 2.

Isotope fractionation is due to small differences in mass and thermodynamic properties of molecules with different isotopic substitution(s) (i.e., isotopologues). Heavier molecules generally have a lower mobility and lower zero-point energy. In the case of water this leads to a higher binding energy and lower vapor pressure of the heavier isotopologues. In the isotope literature the resulting fractionation is often referred to as the vapor pressure isotope effect in (see, e.g.: Bigeleisen, J. 1961. "Statistical Mechanics of Isotope Effects on the Thermodynamic Properties of Condensed Systems." The Journal of Chemical Physics 34 (5): 1485–93. https://doi.org/10.1063/1.1701033, Hook, W.A. Van. 1968. "Vapor Pressures of the Isotopic Waters and Ices." The Journal of Physical Chemistry 72 (4): 1234–44. https://doi.org/10.1021/j100850a028, and Höpfner, A. 1969. "Vapor Pressure Isotope Effects." Angewandte Chemie International Edition in English 8 (10): 689–99. https://doi.org/10.1002/anie.196906891,). For more recent introductions to the subject of fractionation I would like to refer to the excellent book by Zachary Sharp (available at https://digitalrepository.unm.edu/unm_oer/1/) or the series by Wim Mook published by IAEA-UNESCO (available at https://www.hydrology.nl/ihppublications/149-environmental-isotopes-in-the-hydrological-cycle-principles-and-applications.html) . I am unfortunately not very familiar with the phenomena of convective heat transfer in

terms of the Nusselt and Sherwood numbers.

Lines 35 to 40: How does the Author know the isotope composition at any one time during the instrument's start-up, steady-state equilibrium, and variations in steady-state operation e.g. due to changes in droplet size. How was this analysed? What is the frequency or need for isotope re-calibration?

In order to be able to know the isotope composition of the water throughout the system (from the syringe bulk water to the water vapor following evaporation from the droplet surface) it is necessary and sufficient to leave the system running long enough to reach steady-state. As the model analysis and the data show, this typically takes several minutes. The quantitative behavior during the dynamic transitions between different steady states is the subject of the analysis of this paper. It was possible to measure these variations because the laser isotope analyzer used in this study has a 2-Hz measurement update frequency, and a response time limited by a memory effect due to surface adsorption of water to the exposed internal surfaces of the inlet system and the measurement cavity. This response time is of the order of ten seconds, as illustrated in Figure 2, and thus sufficiently fast to allow for an accurate characterization of the dynamic behavior of the droplet evaporation. The question concerning the need for re-calibration was addressed here above with the Reviewer's first question.

Line 43: Does the needle tip have a bevelled tip as most Hamilton syringes do, as this would affect the shape, mass and consequently the transport properties of the suspended droplet. Was the needle tip profile engineered in any way?

The needle used in this case was indeed beveled, with the droplet suspended on top (i.e., the normal to the bevel surface is pointing upward). In this configuration, the droplet is closest in terms of the geometrical configuration to a sessile droplet, as studied by other authors [Sefiane, K., S. K. Wilson, S. David, G. J. Dunn, and B. R. Duffy. 2009. "On the Effect of the Atmosphere on the Evaporation of Sessile Droplets of Water." Physics of Fluids 21 (6): 1–31; Erbil, H. 2012. "Evaporation of Pure Liquid

Sessile and Spherical Suspended Drops: A Review." Advances in Colloid and Interface Science 170 (1–2): 67–86]. A photograph is shown in the thesis by Landsberg (2014) and was reproduced in the companion paper by Leroy-Dos Santos (2020). Although it is true that the shape of the droplet will affect the absolute evaporation rate by changing the effective surface area available for evaporation, our results indicate that the simplification concerning the droplet shape in our model is qualitatively and quantitatively satisfactory. If anything, our observation of a slightly lower evaporation rate than found by Walton (2004), and also Sefiane (2009), is in agreement with the experimental sessile droplet having a somewhat smaller surface to volume ratio than the one that is modeled (see the discussion of section 4.4). This observation will be included in the manuscript. It would have been, and in fact still will be, interesting to study the effect of rotating the needle, or to suspend the droplet from a downward pointing, perpendicularly cut needle, as was done in the work by Walton.

Line 60 and 70: To what extent has the Author considered the model's response in terms of heat and mass transfer, to the following. . . o Changes in droplet temperature. o Molecular diffusion, convection, and conduction via the needle to and from the droplet. o Surface tension and surface energy. . . at 0.1mm in diameter these energies must be high. o Is there any isotopic absorption-desorption equilibrium at the evaporation chamber wall? o Are there any static effects that need to be considered? o The droplets are hemispherical in geometry; would a spherical droplet experience enhanced heat and mass transfer due to droplet instability such as oscillations or distortions in the chamber airflow? o What type of water did the Author use. . . distilled, degassed, isotope enriched? o How is the evaporation chamber temperature is maintained? What are the tolerances?

As the reviewer noticed, the paper does not consider questions of heat transfer occurring during the evaporation of the droplet. The underlying principle of the model is conservation of mass of the individual isotopic components. The model input parameter that depends on heat transfer is the evaporation rate; it was estimated by matching

the model's humidity response to liquid water flow variations to the experimental observations. The thus experimentally determined evaporation rate was subsequently compared to values reported in the literature for droplet evaporation under comparable conditions. Some of the cited literature studies did attempt to relate observations to heat transfer and surface tension properties of the droplet [Sefiane 2009, Walton 2004]. This, however, was considered to be beyond the scope of this paper. Water molecules are notorious for sticking to the walls of the apparatus. Collisions with molecules in the air flow will lead to the inverse process of desorption. In steady-state the two processes are in equilibrium and the walls are coated with a thin layer of water molecules. In our experiment, such surface adsorption/desorption leads to a memory effect: old water will mix with new water and this becomes visible during humidity changes of the air flow, and if, and as long as, the two differ in isotopic composition. This is the underlying reason for the (reduced) time response of the water isotope spectrometer. This effect has been minimized by the application of a hydrophobic coating to the exposed surfaces, and notably those of the evaporation chamber. The resulting time response of the water isotope spectrometer is about one order of magnitude shorter than that of the dynamics of the humidity generator. Still, as explained in the manuscript, this effect has been taken into account by convolution of the simulated behavior with the transfer function (impulse time response) of the spectrometer. Droplet oscillations were not observed in our experiment, and given the relatively high frequency (several Hz at least) expected for such oscillation in the very small droplets studied here (Walton 2009), they would in any case not be directly observable given the time response of the spectrometer. The liquid distilled water sample used for the experiments was a laboratory standard of the Center for Isotope Research (CIO) of the University of Groningen (The Netherlands) with isotope ratios close to those of local tap water (slightly depleted with respect to the international VSMOW standard material). It was kept in a sealed bottle for about 5 years before use. Despite careful storage, the fact that the bottle had been opened previously, meant that its absolute isotopic composition could no longer be guaranteed with the precision specified by the CIO (by repeated isotope ratio mass spectrometric

analyses). It was therefore deemed no longer useful for absolute calibration purposes. In the experiments of this paper, only relative isotope ratios are reported, with the laser spectrometer calibrated against the same liquid water used for the droplet evaporation experiment. The absolute isotope ratios are therefore not relevant. In any case, the drift of the sample was estimated to be less than 1 permil for deltaD and less than 0.2 permil for delta18O (due to possible Rayleigh distillation). The evaporation chamber and the remaining parts of the instrument were maintained at 35 Celsius by active temperature stabilization to better than 0.1 degree using a TEC heater-cooling device. This is much better than required for the stability of the instrument, as discussed in the companion paper (Leroy-Dos Santos 2020). In fact, the new device described in that paper relies on passive temperature stabilization in a laboratory environment with a stability of just 1 degree C.

Line 124: The following statement raises questions about the reliability of the data produced by the instrument. . . "a home-built, low-humidity water isotope spectrometer".

The instrument in question was a research instrument designed, and shown [Landsberg 2014], to surpass the performance of commercially available spectrometers (such as those of LGR or Picarro), notably in terms of precision at low levels of humidity (less than 2000 ppmv water in air). The commercial instruments being designed to measure at normal atmospheric conditions with a water concentration around 20,000 ppmv (parts-per-million-per-volume), their measurement precision quickly deteriorates at the humidity levels probed here. In the final manuscript I propose to replace "home-built" by "high-precision".

Line 136: Similarly, the following statement raises questions of reliability in the calibration process e.g. do these standards deteriorate with time? . . . "The standard waters used were left-over working standards".

Please see my response above (Lines 60-70). I propose to eliminate "left-over" from the sentence and explain the origin of the water standards. The reason I added "left-over"

was certainly not to induce doubts about the quality of the standards or the calibration procedure, but rather to not be accused of using expensive water isotope reference materials for a non-critical application. As explained above, since the isotope spectrometer is calibrated using the same water, and thus all measurements are relative deviations, the absolute values of the isotope ratios of the water sample are not truly relevant.

Line 160 and 167: Table 2, data evaluated at 35oC. Is it valid for the Author to use this data? What is the modelling temperature and operational temperature of the device?

The Reviewer has caught a serious omission in the manuscript: it is implied, but nowhere clearly stated that the relevant parts of the instrument are maintained at 35 degrees Celsius. This will be corrected in the final paper. The modeling was carried out with the fractionation factors (input parameters) valid for 35 degrees Celsius in order to match the temperature of the air stream and water flow inside the evaporation chamber.

Line 186: The Author uses the term "useless results"... if used to validate data, whether in a positive or negative sense, the data can hardly be described as useless!

"Useless" was put between accolades to stress that these unexpected results might have been discarded after the problem with the instrument was identified and corrected. But the Reviewer has a point... I propose to write instead: "But these results that otherwise might have been discarded are reported here because . . .".

Please also note the supplement to this comment:
https://amt.copernicus.org/preprints/amt-2020-428/amt-2020-428-AC1-supplement.pdf

---

## Author Comment (AC2) · 16 Mar 2021

Reply to the Interactive comment by anonymous Referee #2 on the manuscript "Modeling the dynamic behavior of a droplet evaporation device for the delivery of isotopically calibrated low-humidity water vapor" (https://doi.org/10.5194/amt-2020-428).

I would like to start with thanking the reviewer (in fact: both reviewers) for the constructive comments that will undoubtedly make for a much better paper. As outlined here below, these will be implemented in the final manuscript where applicable. Here below I will copy the text of the review in black text, then provide my reply in blue, indented

text, immediately following each item of the review (see supplement).

The author presents a mathematical model to calculate the water isotope ratio which has been measured with a new developed calibration instrument described recently in another work published by Leroy-Dos Santos et al. (2020) in AMTD. The results of the presented simulations fit quite well with the measurements of that particular calibration device. I think it would have been more suitable to publish this work as an add-on to the published paper mentioned above. I don't see the unique selling point of this work and the added value of it for the community. Therefore I suggest a major revision.

We (i.e., the authors involved in the Leroy-Dos Santos paper and the current manuscript) have collectively chosen to submit the modeling effort in a companion paper to the experimental paper by Leroy-Dos Santos et al. (2020) for a number of reasons: Firstly, the theoretical understanding of the dynamic behavior has enabled the identification of the droplet evaporation device as an independent tool to investigate isotope fractionation factors involved in liquid-vapor transitions, as well as isotope fractionation occurring during the process of evaporation of cloud water droplets. The same is true for the determination of the evaporation rate of nano- and micro-liter-sized droplets, which has been the subject of a large body of research, starting with the fundamental work of Maxwell and Langmuir, and more recently in the fields of drying, painting and patterning technologies, dehumidification, cooling technologies, desalination, DNA synthesis, etc. In our opinion, this alone merits separate publications in order to reach as many of the potentially interested researchers as possible. Secondly, we need to mention that the actual instrument used to produce the data for the current paper is different from the one presented in the companion paper. The simulations have not been performed to fit the data presented by Leroy-Dos Santos et al. ... In fact, the data used here in section 3.2 for confrontation with the model could not even be produced with the new instrument, as it was expressly (re-) engineered to eliminate the non-ideal behavior that allowed demonstrating the success of the model in simulating the instrument behavior under very different operating conditions. It was also

never equipped with evaporation chambers of different dimensions, as was needed to produce the distinctly different behavior of section 3.1 versus section 3.2. Partly as a consequence of this, a different combination of authors contributed to the work described in the two papers. Finally, we also felt that adding the model to the experimental paper would make the paper unwieldly long. Also, it would have been very awkward, and most likely confusing for the reader, having to describe the older, prototype instrument in the same paper that describes the newly designed, re-engineered device in order to be able to present the data of section 3.2. In conclusion, and especially considering that the experimental paper by Leroy-Dos Santos and colleagues (2020) has now been accepted for publication, and its final version has been submitted, we remain of the opinion that the model paper should be published separately, and preferably as a companion paper.

In any case I have several comments the author should consider to incorporate into the manuscript before publication. Major issues: 1) For a better understanding the author should briefly describe in the introduction why the measurement of the water isotope is important and what insights you gain from it.

I thank the reviewer for this suggestion, as it will indeed be useful for a reader who is not from the isotope community to see some justification for the importance accorded to water isotope measurements. I thus suggest to add a more general explanation at the beginning of the Introduction along the lines of: "Water is arguably the most important molecule in Earth's atmosphere. The large enthalpy change associated with the evaporation and condensation of water causes it to dominate the global redistribution of energy by tropospheric transport of latent heat. Water vapor is also the most important greenhouse gas. The natural atmospheric greenhouse effect warms Earth's surface by 33 K to hospitable temperatures of on average 15 °C. About 75% of this temperature increase is generated by water vapor and clouds, as a feedback effect driven by the non-condensable greenhouse agents, and foremost carbon dioxide [Lacis2010]. This feedback effect, in turn, is a superposition of a multitude of large and (especially

as clouds are involved) complex individual processes that partially cancel each other. Due to this complexity, water, in the form of water vapor, as well as liquid and crystal phase water inside clouds, is by far the largest unknown in current climate models [IPCC2014]. Atmospheric data of relevant tracers, which may help to disentangle and quantify the many relevant processes, are desperately needed. Of these, the isotopic composition of water is arguably the best candidate (in particular the isotope ratios D/H and 18O/16O), as all processes in which water is involved are isotope-dependent. Therefore, water isotope ratios enable identification of different moist air masses and following of their mixing; they also reflect the evaporation and condensation history of the moist air in question. In the journal Nature, the climate researcher Gavin Schmidt actually called the water isotopes "the most super-duper fantastic thing ever" [Tollefson2008]." This can then be followed by the specific argumentation, as we put forward in the Leroy-Dos Santos paper that there is a strong need for a reliable means of calibrating optical spectrometers for low-humidity applications, as in Antarctica, using liquid standards.

Tollefson, Jeff. 2008. "Vapour Spies to Reveal Climate Clues." Nature 455 (7214): 714–714. https://doi.org/10.1038/455714a.

2) It would be good to describe the meaning of $\delta$ for the readers not so familiar with the subject.

Thank you for pointing this out. I will include a brief description in the Introduction of the delta-value as a relative deviation of the isotope ratio in a sample with respect to the same ratio in a standard material. The definition of the delta-value should appear much earlier in the paper than at L100, as is now the case.

3) The derivation of the formulas are not always easy to follow, please prepare a better description.

It is disappointing to read this as I had the manuscript already proofread by more than one colleague, and had made modifications following their suggestions . . . In the absence of more detailed comments, I will re-read the entire paper again with special attention to this issue, and in the process ask for additional input from different colleagues to be incorporated in the final version. At this point I can think of two clarifications that may be useful to some readers: L89: Add an intermediate step: "expressing both the droplet size . . . and the evaporative water flux Phi_e(t) as a function of V_d(t), and to subsequently calculate both as function of the time-dependent input water flux Phi_0(t) by numerical integration of Eq. (3)." L90-100: Mention that the factor (1+R) in the denominator of Eqs. (9) to (11) arises from a conversion from isotope ratio to isotope concentration (requiring that one divides by the sum of the isotopologues, not just the abundant isotopologue).

4) Generally please replace 'water concentration' with 'water vapour concentration'..

This will be done. See also the comment on this topic by David Walton (Referee 1).

5) All figures should be enlarged. The legend of the figures are too small..

Thank you for this suggestion; will be implemented in the final version. Please note that the current type setting is imposed by the TeX style document provided by AMT. The definition of the images is more than sufficient to allow for larger figures. For the manuscript I set the figure sizes manually, but I expect that these will be overruled in the journal's final type setting process.

6) The colours of the graphs changes from one figure to the next. The author should consider to use the same colour for the simulation in all figures.

Thank you for pointing this out. Whereas the color coding: Black for concentrations, Red for Oxygen-18, and Green for Deuterium has been followed throughout, Figures 4 and 6 should have their color coding inversed with respect to experimental data (should be gray) and simulation (one of the above colors for the isotopes). This will be easy to correct for the final version.

7) Figure 4 why does $\delta$2H oscillate so much more than $\delta$18O. Please give an explanation.

Thank you for this observation. I will add an explanation to the text to mention that the larger fractionation for deuterium compared to oxygen-18 leads to a larger excursion of the d2H signal compared to d18O. At the same time the noise on the d2H signal is much larger than that of d18O because of the lower signal-to-noise ratio of the deuterium feature detected by the infrared spectrometer, directly related to the lower abundance of 2H16O1H (0.031%) with respect to 1H18O1H (0.20%) in our natural water sample.

Minor issues: 1) line 66-69 the use of numbers in brackets could lead to the misunderstanding that the author refers to the number of the appropriate formulas, consider using other annotations. 2) line 136 The standard waters used were left-over? Consider a better formulation.

Thank you for pointing out the potential for misreading the numbers in lines 66-69. I will itemize the three points using the letters a, b, and c instead. Considering the "left-over" in line 136, I refer to my response to Dr. Walton's question concerning the type of water used (with Lines 60-70). I propose to mention the origin of the water as an older standard water of the Center for Isotope Research in The Netherlands.

Please also note the supplement to this comment:
https://amt.copernicus.org/preprints/amt-2020-428/amt-2020-428-AC2-supplement.pdf

---

## Author Response (AR2)

**MANUSCRIPT AMT-2020-428-version5: technical corrections**
Modeling the dynamic behavior of a droplet evaporation device for the delivery of isotopically calibrated low-humidity water vapor
Erik Kerstel

Grenoble, May 18, 2021

Dear Marc,

Thank you very much for your handling of my manuscript!

 Both technical corrections demanded in the 2$^{nd}$ reviewing process have been addressed in this latest version of the manuscript:

Line 46: replaced "it" by "its".
Figure 4: Modified the caption to add "(gray curves)" and "(green for δ2H and red for δ18O)".
Figure 5: Similar to Figure 5, added an explanation of the color coding to the caption.

Very best regards, Erik